# From Stochastic Mixability to Fast Rates

**Nishant A. Mehta**
Research School of Computer Science
Australian National University
nishant.mehta@anu.edu.au

**Robert C. Williamson**
Research School of Computer Science
Australian National University and NICTA
bob.williamson@anu.edu.au

## Abstract

Empirical risk minimization (ERM) is a fundamental learning rule for statistical learning problems where the data is generated according to some unknown distribution $\mathsf{P}$ and returns a hypothesis $f$ chosen from a fixed class $\mathcal{F}$ with small loss $\ell$. In the parametric setting, depending upon $(\ell, \mathcal{F}, \mathsf{P})$ ERM can have slow $(1/\sqrt{n})$ or fast $(1/n)$ rates of convergence of the excess risk as a function of the sample size $n$. There exist several results that give sufficient conditions for fast rates in terms of joint properties of $\ell$, $\mathcal{F}$, and $\mathsf{P}$, such as the margin condition and the Bernstein condition. In the non-statistical prediction with expert advice setting, there is an analogous slow and fast rate phenomenon, and it is entirely characterized in terms of the mixability of the loss $\ell$ (there being no role there for $\mathcal{F}$ or $\mathsf{P}$). The notion of stochastic mixability builds a bridge between these two models of learning, reducing to classical mixability in a special case. The present paper presents a direct proof of fast rates for ERM in terms of stochastic mixability of $(\ell, \mathcal{F}, \mathsf{P})$, and in so doing provides new insight into the fast-rates phenomenon. The proof exploits an old result of Kemperman on the solution to the general moment problem. We also show a partial converse that suggests a characterization of fast rates for ERM in terms of stochastic mixability is possible.

## 1  Introduction

Recent years have unveiled central contact points between the areas of statistical and online learning. These include Abernethy et al.'s [1] unified Bregman-divergence based analysis of online convex optimization and statistical learning, the online-to-batch conversion of the exponentially weighted average forecaster (a special case of the aggregating algorithm for mixable losses) which yields the progressive mixture rule as can be seen e.g. from the work of Audibert [2], and most recently Van Erven et al.'s [21] injection of the concept of mixability into the statistical learning space in the form of stochastic mixability. It is this last connection that will be our departure point for this work.

Mixability is a fundamental property of a loss that characterizes when constant regret is possible in the online learning game of prediction with expert advice [23]. Stochastic mixability is a natural adaptation of mixability to the statistical learning setting; in fact, in the special case where the function class consists of all possible functions from the input space to the prediction space, stochastic mixability is equivalent to mixability [21]. Just as Vovk and coworkers (see e.g. [24, 8]) have developed a rich convex geometric understanding of mixability, stochastic mixability can be understood as a sort of effective convexity.

In this work, we study the $O(1/n)$-fast rate phenomenon in statistical learning from the perspective of stochastic mixability. Our motivation is that stochastic mixability might characterize fast rates in statistical learning. As a first step, Theorem 5 herein establishes via a rather direct argument that stochastic mixability implies an exact oracle inequality (i.e. with leading constant 1) with a fast rate for finite function classes, and Theorem 7 extends this result to VC-type classes. This result can be understood as a new chapter in an evolving narrative that started with Lee et al.'s [13] seminal paper

showing fast rates for agnostic learning with squared loss over convex function classes, and that was continued by Mendelson [18] who showed that fast rates are possible for $p$-losses $(y, \hat{y}) \mapsto |y - \hat{y}|^p$ over effectively convex function classes by passing through a Bernstein condition (defined in (12)).

We also show that when stochastic mixability does not hold in a certain sense (described in Section 5), then the risk minimizer is not unique in a bad way. This is precisely the situation at the heart of the works of Mendelson [18] and Mendelson and Williamson [19], which show that having non-unique minimizers is symptomatic of bad geometry of the learning problem. In such situations, there are certain targets (i.e. output conditional distributions) close to the original target under which empirical risk minimization learns (ERM) at a slow rate, where the guilty target depends on the sample size and the target sequence approaches the original target asymptotically. Even the best known upper bounds have constants that blow up in the case of non-unique minimizers. Thus, whereas stochastic mixability implies fast rates, a sort of converse is also true, where learning is hard in a "neighborhood" of statistical learning problems for which stochastic mixability does not hold. In addition, since a stochastically mixable problem's function class looks convex from the perspective of risk minimization, and since when stochastic mixability fails the function class looks non-convex from the same perspective (it has multiple well-separated minimizers), stochastic mixability *characterizes* the effective convexity of the learning problem from the perspective of risk minimization.

Much of the recent work in obtaining faster learning rates in agnostic learning has taken place in settings where a Bernstein condition holds, including results based on local Rademacher complexities [3, 10]. The Bernstein condition appears to have first been used by Bartlett and Mendelson [4] in their analysis of ERM; this condition is subtly different from the margin condition of Mammen and Tsybakov [15, 20], which has been used to obtain fast rates for classification. Lecué [12] pinpoints that the difference between the two conditions is that the margin condition applies to the excess loss relative to the best predictor (not necessarily in the model class) whereas the Bernstein condition applies to the excess loss relative to the best predictor in the model class. Our approach in this work is complementary to the approaches of previous works, coming from a different assumption that forms a bridge to the online learning setting. Yet this assumption is related; the Bernstein condition implies stochastic mixability under a bounded losses assumption [21]. Further understanding the connection between the Bernstein condition and stochastic mixability is an ongoing effort.

**Contributions.** The core contribution of this work is to show a new path to the $\tilde{O}(1/n)$-fast rate in statistical learning. We are not aware of previous results that show fast rates from the stochastic mixability assumption. Secondly, we establish intermediate learning rates that interpolate between the fast and slow rate under a weaker notion of stochastic mixability. Finally, we show that in a certain sense stochastic mixability characterizes the effective convexity of the statistical problem.

In the next section we formally define the statistical problem, review stochastic mixability, and explain our high-level approach toward getting fast rates. This approach involves directly appealing to the Cramér-Chernoff method, from which nearly all known concentration inequalities arose in one way or another. In Section 3, we frame the problem of computing a particular moment of a certain excess loss random variable as a general moment problem. We sufficiently bound the optimal value of the moment, which allows for a direct application of the Cramér-Chernoff method. These results easily imply a fast rates bound for finite classes that can be extended to parametric (VC-type) classes, as shown in Section 4. We describe in Section 5 how stochastic mixability characterizes a certain notion of convexity of the statistical learning problem. In Section 6, we extend the fast rates results to classes that obey a notion we call weak stochastic mixability. Finally, Section 7 concludes this work with connections to related topics in statistical learning theory and a discussion of open problems.

## 2 Stochastic mixability, Cramér-Chernoff, and ERM

Let $(\ell, \mathcal{F}, \mathsf{P})$ be a statistical learning problem with $\ell : \mathcal{Y} \times \mathbb{R} \to \mathbb{R}_+$ a nonnegative loss, $\mathcal{F} \subset \mathbb{R}^{\mathcal{X}}$ a compact function class, and $\mathsf{P}$ a probability measure over $\mathcal{X} \times \mathcal{Y}$ for input space $\mathcal{X}$ and output/target space $\mathcal{Y}$. Let $Z$ be a random variable defined as $Z = (X, Y) \sim \mathsf{P}$. We assume for all $f \in \mathcal{F}$, $\ell(Y, f(X)) \leq V$ almost surely (a.s.) for some constant $V$.

A probability measure $\mathsf{P}$ operates on functions and loss-composed functions as:

$$\mathsf{P} f = \mathsf{E}_{(X,Y) \sim \mathsf{P}} f(X) \qquad \qquad \mathsf{P} \ell(\cdot, f) = \mathsf{E}_{(X,Y) \sim \mathsf{P}} \ell\big(Y, f(X)\big).$$

Similarly, an empirical measure $\mathsf{P}_n$ associated with an $n$-sample $\mathbf{z}$, comprising $n$ iid samples $(x_1, y_1), \ldots, (x_n, y_n)$, operates on functions and loss-composed functions as:

$$\mathsf{P}_n f = \frac{1}{n} \sum_{j=1}^n f(x_j) \qquad\qquad \mathsf{P}_n \ell(\cdot, f) = \frac{1}{n} \sum_{j=1}^n \ell\big(y_j, f(x_j)\big).$$

Let $f^*$ be any function for which $\mathsf{P}\,\ell(\cdot, f^*) = \inf_{f \in \mathcal{F}} \mathsf{P}\,\ell(\cdot, f)$. For each $f \in \mathcal{F}$ define the excess risk random variable $Z_f := \ell\big(Y, f(X)\big) - \ell\big(Y, f^*(X)\big)$.

We frequently work with the following two subclasses. For any $\varepsilon > 0$, define the subclasses
$$\mathcal{F}_{\preceq \varepsilon} := \{ f \in \mathcal{F} : \mathsf{P}\, Z_f \leq \varepsilon \} \qquad\qquad \mathcal{F}_{\succeq \varepsilon} := \{ f \in \mathcal{F} : \mathsf{P}\, Z_f \geq \varepsilon \} .$$

## 2.1 Stochastic mixability

For $\eta > 0$, we say that $(\ell, \mathcal{F}, \mathsf{P})$ is $\eta$-*stochastically mixable* if for all $f \in \mathcal{F}$
$$\log \mathsf{E} \exp(-\eta Z_f) \leq 0. \tag{1}$$
If $\eta$-stochastic mixability holds for some $\eta > 0$, then we say that $(\ell, \mathcal{F}, \mathsf{P})$ is *stochastically mixable*. Throughout this paper it is assumed that the stochastic mixability condition holds, and we take $\eta^*$ to be the largest $\eta$ such that $\eta$-stochastic mixability holds. Condition (1) has a rich history, beginning from the foundational thesis of Li [14] who studied the special case of $\eta^* = 1$ in density estimation with log loss from the perspective of information geometry. The connections that Li showed between this condition and convexity were strengthened by Grünwald [6, 7] and Van Erven et al. [21].

## 2.2 Cramér-Chernoff

The high-level strategy taken here is to show that with high probability ERM will not select a fixed hypothesis function $f$ with excess risk above $\frac{a}{n}$ for some constant $a > 0$. For each hypothesis, this guarantee will flow from the Cramér-Chernoff method [5] by controlling the cumulant generating function (CGF) of $-Z_f$ in a particular way to yield exponential concentration. This control will be possible because the $\eta^*$-stochastic mixability condition implies that the CGF of $-Z_f$ takes the value 0 at some $\eta \geq \eta^*$, a fact later exploited by our key tool Theorem 3.

Let $Z$ be a real-valued random variable. Applying Markov's inequality to an exponentially transformed random variable yields that, for any $\eta \geq 0$ and $t \in \mathbb{R}$
$$\mathsf{Pr}(Z \geq t) \leq \exp(-\eta t + \log \mathsf{E} \exp(\eta Z)); \tag{2}$$
the inequality is non-trivial only if $t > \mathsf{E}\, Z$ and $\eta > 0$.

## 2.3 Analysis of ERM

We consider the *ERM estimator* $\hat{f}_{\mathbf{z}} := \arg\min_{f \in \mathcal{F}} \mathsf{P}_n \ell(\cdot, f)$. That is, given an $n$-sample $\mathbf{z}$, ERM selects any $\hat{f}_{\mathbf{z}} \in \mathcal{F}$ minimizing the empirical risk $\mathsf{P}_n \ell(\cdot, f)$. We say ERM is $\varepsilon$-*good* when $\hat{f}_{\mathbf{z}} \in \mathcal{F}_{\preceq \varepsilon}$. In order to show that ERM is $\varepsilon$-good it is sufficient to show that for all $f \in \mathcal{F} \setminus \mathcal{F}_{\preceq \varepsilon}$ we have $\mathsf{P}\, Z_f > 0$. The goal is to show that with high probability ERM is $\varepsilon$-good, and we will do this by showing that with high probability uniformly for all $f \in \mathcal{F} \setminus \mathcal{F}_{\preceq \varepsilon}$ we have $\mathsf{P}_n Z_f > t$ for some slack $t > 0$ that will come in handy later.

For a real-valued random variable $X$, recall that the *cumulant generating function* of $X$ is $\eta \mapsto \Lambda_X(\eta) := \log \mathsf{E}\, e^{\eta X}$; we allow $\Lambda_X(\eta)$ to be infinite for some $\eta > 0$.

**Theorem 1 (Cramér-Chernoff Control on ERM).** *Let $a > 0$ and select $f$ such that $\mathsf{E}\, Z_f > 0$. Let $t < \mathsf{E}\, Z_f$. If there exists $\eta > 0$ such that $\Lambda_{-Z_f}(\eta) \leq -\frac{a}{n}$, then*
$$\mathsf{Pr}\Big\{ \mathsf{P}_n \ell(\cdot, f) \leq \mathsf{P}_n \ell(\cdot, f^*) + t \Big\} \leq \exp(-a + \eta t).$$

*Proof.* Let $Z_{f,1}, \ldots, Z_{f,n}$ be iid copies of $Z_f$, and define the sum $S_{f,n} := \sum_{j=1}^n -Z_{f,j}$. Since $(-t) > \mathsf{E}\, \frac{1}{n} S_{f,n}$, then from (2) we have
$$\mathsf{Pr}\left( \frac{1}{n} \sum_{j=1}^n Z_{f,j} \leq t \right) = \mathsf{Pr}\left( \frac{1}{n} S_{f,n} \geq -t \right) \leq \exp\left( \eta t + \log \mathsf{E} \exp(\eta S_{f,n}) \right)$$
$$= \exp(\eta t)\big( \mathsf{E} \exp(-\eta Z_f) \big)^n.$$

Making the replacement $\Lambda_{-Z_f}(\eta) = \log \mathsf{E}\exp(-\eta Z_f)$ yields

$$\log \mathsf{Pr}\left(\frac{1}{n}S_{f,n} \geq -t\right) \leq \eta t + n\Lambda_{-Z_f}(\eta).$$

By assumption, $\Lambda_{-Z_f}(\eta) \leq -\frac{a}{n}$, and so $\mathsf{Pr}\{\mathsf{P}_n Z_f \leq t\} \leq \exp(-a + \eta t)$ as desired. $\qquad\square$

This theorem will be applied by showing that for an excess loss random variable $Z_f$ taking values in $[-1, 1]$, if for some $\eta > 0$ we have $\mathsf{E}\exp(-\eta Z_f) = 1$ and if $\mathsf{E} Z_f = \frac{a}{n}$ for some constant $a$ (that can and must depend on $n$), then $\Lambda_{-Z_f}(\eta/2) \leq -\frac{c\eta a}{n}$ where $c > 0$ is a universal constant. This is the nature of the next section. We then extend this result to random variables taking values in $[-V, V]$.

## 3   Semi-infinite linear programming and the general moment problem

The key subproblem now is to find, for each excess loss random variable $Z_f$ with mean $\frac{a}{n}$ and $\Lambda_{-Z_f}(\eta) = 0$ (for some $\eta \geq \eta^*$), a pair of constants $\eta_0 > 0$ and $c > 0$ for which $\Lambda_{-Z_f}(\eta_0) \leq -\frac{ca}{n}$. Theorem 1 would then imply that ERM will prefer $f^*$ over this particular $f$ with high probability for $ca$ large enough. This subproblem is in fact an instance of the general moment problem, a problem on which Kemperman [9] has conducted a very nice geometric study. We now describe this problem.

**The general moment problem.**   Let $\mathcal{P}(\mathcal{A})$ be the space of probability measures over a measurable space $\mathcal{A} = (\mathcal{A}, \mathcal{S})$. For real-valued measurable functions $h$ and $(g_j)_{j\in[m]}$ on a measurable space $\mathcal{A} = (\mathcal{A}, \mathcal{S})$, the general moment problem is

$$\begin{aligned}
&\inf_{\mu\in\mathcal{P}(\mathcal{A})} \quad \mathsf{E}_{X\sim\mu}\, h(X) \\
&\text{subject to} \quad \mathsf{E}_{X\sim\mu}\, g_j(X) = y_j, \quad j \in \{1,\ldots,m\}.
\end{aligned} \tag{3}$$

Let the vector-valued map $g : \mathcal{A} \to \mathbb{R}^m$ be defined in terms of coordinate functions as $(g(x))_j = g_j(x)$, and let the vector $y \in \mathbb{R}^m$ be equal to $(y_1, \ldots, y_m)$.

Let $D^* \subset \mathbb{R}^{m+1}$ be the set

$$D^* := \left\{ d^* = (d_0, d_1, \ldots, d_m) \in \mathbb{R}^{m+1} : h(x) \geq d_0 + \sum_{j=1}^m d_j g_j(x) \quad \text{for all } x \in \mathcal{A} \right\}. \tag{4}$$

Theorem 3 of [9] states that if $y \in \operatorname{int}\operatorname{conv} g(\mathcal{A})$, the optimal value of problem (3) equals

$$\sup\left\{ d_0 + \sum_{j=1}^m d_j y_j : d^* = (d_0, d_1, \ldots, d_m) \in D^* \right\}. \tag{5}$$

**Our instantiation.**   We choose $\mathcal{A} = [-1, 1]$, set $m = 2$ and define $h$, $(g_j)_{j\in\{1,2\}}$, and $y \in \mathbb{R}^2$ as:

$$h(x) = -e^{(\eta/2)x}, \qquad g_1(x) = x, \qquad g_2(x) = e^{\eta x}, \qquad y_1 = -\frac{a}{n}, \qquad y_2 = 1,$$

for any $\eta > 0$, $a > 0$, and $n \in \mathbb{N}$. This yields the following instantiation of problem (3):

$$\inf_{\mu\in\mathcal{P}([-1,1])} \quad \mathsf{E}_{X\sim\mu} -e^{(\eta/2)X} \tag{6a}$$

$$\text{subject to} \quad \mathsf{E}_{X\sim\mu} X = -\frac{a}{n} \tag{6b}$$

$$\mathsf{E}_{X\sim\mu} e^{\eta X} = 1. \tag{6c}$$

Note that equation (5) from the general moment problem now instantiates to

$$\sup\left\{ d_0 - \frac{a}{n}d_1 + d_2 : d^* = (d_0, d_1, d_2) \in D^* \right\}, \tag{7}$$

with $D^*$ equal to the set

$$\left\{ d^* = (d_0, d_1, d_2) \in \mathbb{R}^3 : -e^{(\eta/2)x} \geq d_0 + d_1 x + d_2 e^{\eta x} \quad \text{for all } x \in [-1, 1] \right\}. \tag{8}$$

Applying Theorem 3 of [9] requires the condition $y \in \operatorname{int}\operatorname{conv} g([-1, 1])$. We first characterize when $y \in \operatorname{conv} g([-1, 1])$ holds and handle the $\operatorname{int}\operatorname{conv} g([-1, 1])$ version after Theorem 3.

**Lemma 2 (Feasible Moments).** *The point* $y = \left(-\frac{a}{n}, 1\right) \in \operatorname{conv} g([-1,1])$ *if and only if*

$$\frac{a}{n} \leq \frac{e^{\eta} + e^{-\eta} - 2}{e^{\eta} - e^{-\eta}} = \frac{\cosh(\eta) - 1}{\sinh(\eta)}. \tag{9}$$

*Proof.* Let $W$ denote the convex hull of $g([-1,1])$. We need to see if $\left(-\frac{a}{n}, 1\right) \in W$. Note that $W$ is the convex set formed by starting with the graph of $x \mapsto e^{\eta x}$ on the domain $[-1,1]$, including the line segment connecting this curve's endpoints $(-1, e^{-\eta})$ to $(1, e^{\eta x})$, and including all of the points below this line segment but above the aforementioned graph. That is, $W$ is precisely the set

$$W := \left\{ (x,y) \in \mathbb{R}^2 : e^{\eta x} \leq y \leq \frac{e^{\eta} + e^{-\eta}}{2} + \frac{e^{\eta} - e^{-\eta}}{2} x, \ \forall x \in [-1,1] \right\}.$$

It remains to check that $1$ is sandwiched between the lower and upper bounds at $x = -\frac{a}{n}$. Clearly the lower bound holds. Simple algebra shows that the upper bound is equivalent to condition (9). $\square$

Note that if (9) does not hold, then the semi-infinite linear program (6) is infeasible; infeasibility in turn implies that such an excess loss random variable cannot exist. Thus, we need not worry about whether (9) holds; it holds for *any* excess loss random variable satisfying constraints (6b) and (6c).

The following theorem is a key technical result for using stochastic mixability to control the CGF. The proof is long and can be found in Appendix A.

**Theorem 3 (Stochastic Mixability Concentration).** *Let* $f$ *be an element of* $\mathcal{F}$ *with* $Z_f$ *taking values in* $[-1,1]$, $n \in \mathbb{N}$, $\mathsf{E}\, Z_f = \frac{a}{n}$ *for some* $a > 0$, *and* $\Lambda_{-Z_f}(\eta) = 0$ *for some* $\eta > 0$. *If*

$$\frac{a}{n} < \frac{e^{\eta} + e^{-\eta} - 2}{e^{\eta} - e^{-\eta}}, \tag{10}$$

*then*
$$\mathsf{E}\, e^{(\eta/2)(-Z_f)} \leq 1 - \frac{0.18(\eta \wedge 1)a}{n}.$$

Note that since $\log(1-x) \leq -x$ when $x < 1$, we have $\Lambda_{-Z_f}(\eta/2) \leq -\frac{0.18(\eta \wedge 1)a}{n}$.

In order to apply Theorem 3, we need (10) to hold, but only (9) is guaranteed to hold. The corner case is if (9) holds with equality. However, observe that one can always approximate the random variable $X$ by a perturbed version $X'$ which has nearly identical mean $a' \approx a$ and a nearly identical $\eta' \approx \eta$ for which $\mathsf{E}_{X' \sim \mu'}\, e^{\eta' X'} = 1$, and yet the inequality in (9) is strict. Later, in the proof of Theorem 5, for any random variable that required perturbation to satisfy the interior condition (10), we implicitly apply the analysis to the perturbed version, show that ERM would not pick the (slightly different) function corresponding to the perturbed version, and use the closeness of the two functions to show that ERM also would not pick the original function.

We now present a necessary extension for the case of losses with range $[0, V]$, proved in Appendix A.

**Lemma 4 (Bounded Losses).** *Let* $g_1(x) = x$ *and* $y_2 = 1$ *be common settings for the following two problems. The instantiation of problem* (3) *with* $\mathcal{A} = [-V, V]$, $h(x) = -e^{(\eta/2)x}$, $g_2(x) = e^{\eta x}$, *and* $y_1 = -\frac{a}{n}$ *has the same optimal value as the instantiation of problem* (3) *with* $\mathcal{A} = [-1, 1]$, $h(x) = -e^{(V\eta/2)x}$, $g_2(x) = e^{(V\eta)x}$, *and* $y_1 = -\frac{a/V}{n}$.

## 4 Fast rates

We now show how the above results can be used to obtain an exact oracle inequality with a fast rate. We first present a result for finite classes and then present a result for VC-type classes (classes with logarithmic universal metric entropy).

**Theorem 5 (Finite Classes Exact Oracle Inequality).** *Let* $(\ell, \mathcal{F}, \mathsf{P})$ *be* $\eta^*$-*stochastically mixable, where* $|\mathcal{F}| = N$, $\ell$ *is a nonnegative loss, and* $\sup_{f \in \mathcal{F}} \ell\big(Y, f(X)\big) \leq V$ *a.s. for a constant* $V$. *Then for all* $n \geq 1$, *with probability at least* $1 - \delta$

$$\mathsf{P}\, \ell(\cdot, \hat{f}_{\mathbf{z}}) \leq \mathsf{P}\, \ell(\cdot, f^*) + \frac{6 \max\left\{V, \frac{1}{\eta^*}\right\} \left(\log \frac{1}{\delta} + \log N\right)}{n}.$$

*Proof.* Let $\gamma_n = \frac{a}{n}$ for a constant $a$ to be fixed later. For each $\eta > 0$, let $\mathcal{F}^{(\eta)}_{\succeq \gamma_n} \subset \mathcal{F}_{\succeq \gamma_n}$ correspond to those functions in $\mathcal{F}_{\succeq \gamma_n}$ for which $\eta$ is the largest constant such that $\mathsf{E}\exp(-\eta Z_f) = 1$. Let $\mathcal{F}^{\text{hyper}}_{\succeq \gamma_n} \subset \mathcal{F}_{\succeq \gamma_n}$ correspond to functions $f$ in $\mathcal{F}_{\succeq \gamma_n}$ for which $\lim_{\eta \to \infty} \mathsf{E}\exp(-\eta Z_f) < 1$. Clearly, $\mathcal{F}_{\succeq \gamma_n} = \left( \bigcup_{\eta \in [\eta^*, \infty)} \mathcal{F}^{(\eta)}_{\succeq \gamma_n} \right) \cup \mathcal{F}^{\text{hyper}}_{\succeq \gamma_n}$. The excess loss random variables corresponding to elements $f \in \mathcal{F}^{\text{hyper}}_{\succeq \gamma_n}$ are "hyper-concentrated" in the sense that they are infinitely stochastically mixable. However, Lemma 10 in Appendix B shows that for each hyper-concentrated $Z_f$, there exists another excess loss random variable $Z'_f$ with mean arbitrarily close to that of $Z_f$, with $\mathsf{E}\exp(-\eta Z'_f) = 1$ for some arbitrarily large but finite $\eta$, and with $Z'_f \leq Z_f$ with probability 1. The last property implies that the empirical risk of $Z'_f$ is no greater than that of $Z_f$; hence for each hyper-concentrated $Z_f$ it is sufficient (from the perspective of ERM) to study a corresponding $Z'_f$. From now on, we implicitly make this replacement in $\mathcal{F}_{\succeq \gamma_n}$ itself, so that we now have $\mathcal{F}_{\succeq \gamma_n} = \bigcup_{\eta \in [\eta^*, \infty)} \mathcal{F}^{(\eta)}_{\succeq \gamma_n}$.

Consider an arbitrary $a > 0$. For some fixed $\eta \in [\eta^*, \infty)$ for which $|\mathcal{F}^{(\eta)}_{\succeq \gamma_n}| > 0$, consider the subclass $\mathcal{F}^{(\eta)}_{\succeq \gamma_n}$. Individually for each such function, we will apply Theorem 1 as follows. From Lemma 4, we have $\Lambda_{-Z_f}(\eta/2) = \Lambda_{-\frac{1}{V}Z_f}(V\eta/2)$. From Theorem 3, the latter is at most $-\frac{0.18(V\eta \wedge 1)(a/V)}{n} = -\frac{0.18\eta a}{(V\eta \vee 1)n}$ . Hence, Theorem 1 with $t = 0$ and the $\eta$ from the Theorem taken to be $\eta/2$ implies that the probability of the event $\mathsf{P}_n \ell(\cdot, f) \leq \mathsf{P}_n \ell(\cdot, f^*)$ is at most $\exp\left(-0.18 \frac{\eta}{V\eta \vee 1} a\right)$. Applying the union bound over all of $\mathcal{F}_{\succeq \gamma_n}$, we conclude that

$$\Pr\left\{\exists f \in \mathcal{F}_{\succeq \gamma_n} : \mathsf{P}_n \ell(\cdot, f) \leq \mathsf{P}_n \ell(\cdot, f^*)\right\} \leq N \exp\left(-\eta^* \left(\frac{0.18a}{V\eta^* \vee 1}\right)\right).$$

Since ERM selects hypotheses on their empirical risk, from inversion it holds that with probability at least $1 - \delta$ ERM will not select any hypothesis with excess risk at least $\frac{6\max\left\{V, \frac{1}{\eta^*}\right\}\left(\log\frac{1}{\delta} + \log N\right)}{n}$. $\square$

Before presenting the result for VC-type classes, we require some definitions. For a pseudometric space $(\mathcal{G}, d)$, for any $\varepsilon > 0$, let $\mathcal{N}(\varepsilon, \mathcal{G}, d)$ be the $\varepsilon$-covering number of $(\mathcal{G}, d)$; that is, $\mathcal{N}(\varepsilon, \mathcal{G}, d)$ is the minimal number of balls of radius $\varepsilon$ needed to cover $\mathcal{G}$. We will further constrain the cover (the set of centers of the balls) to be a subset of $\mathcal{G}$ (i.e. to be proper), thus ensuring that the stochastic mixability assumption transfers to any (proper) cover of $\mathcal{F}$. Note that the "proper" requirement at most doubles the constant $K$ below, as shown by Vidyasagar [22, Lemma 2.1].

We now state a localization-based result that allows us to extend the result for finite classes to VC-type classes. Although the localization result can be obtained by combining standard techniques,[1] we could not find this particular result in the literature. Below, an $\varepsilon$-net $\mathcal{F}_\varepsilon$ of a set $\mathcal{F}$ is a subset of $\mathcal{F}$ such that $\mathcal{F}$ is contained in the union of the balls of radius $\varepsilon$ with centers in $\mathcal{F}_\varepsilon$.

**Theorem 6.** *Let $\mathcal{F}$ be a separable function class whose functions have range bounded in $[0, V]$ and for which, for a constant $K \geq 1$, for each $u \in (0, K]$ the $L_2(\mathsf{P})$ covering numbers are bounded as*

$$\mathcal{N}(u, \mathcal{F}, L_2(\mathsf{P})) \leq \left(\frac{K}{u}\right)^{\mathcal{C}}. \tag{11}$$

*Suppose $\mathcal{F}_\varepsilon$ is a minimal $\varepsilon$-net for $\mathcal{F}$ in the $L_2(\mathsf{P})$ norm, with $\varepsilon = \frac{1}{n}$. Denote by $\pi : \mathcal{F} \to \mathcal{F}_\varepsilon$ an $L_2(\mathsf{P})$-metric projection from $\mathcal{F}$ to $\mathcal{F}_\varepsilon$. Then, provided that $\delta \leq \frac{1}{2}$, with probability at most $\delta$ can there exist $f \in \mathcal{F}$ such that*

$$\mathsf{P}_n f < \mathsf{P}_n(\pi(f)) - \frac{V}{n}\left(1080\mathcal{C}\log(2Kn) + 90\sqrt{\left(\log\frac{1}{\delta}\right)\mathcal{C}\log(2Kn)} + \log\frac{e}{\delta}\right).$$

The proof is presented in Appendix C. We now present the fast rates result for VC-type classes. The proof (in Appendix C) uses Theorem 6 and the proof of the Theorem 5. Below, we denote the loss-composed version of a function class $\mathcal{F}$ as $\ell \circ \mathcal{F} := \{\ell(\cdot, f) : f \in \mathcal{F}\}$.

**Theorem 7 (VC-Type Classes Exact Oracle Inequality).** *Let $(\ell, \mathcal{F}, \mathsf{P})$ be $\eta^*$-stochastically mixable with $\ell \circ \mathcal{F}$ separable, where, for a constant $K \geq 1$, for each $\varepsilon \in (0, K]$ we have $\mathcal{N}(\ell \circ \mathcal{F}, L_2(\mathsf{P}), \varepsilon) \leq \left(\frac{K}{\varepsilon}\right)^{\mathcal{C}}$, and $\sup_{f \in \mathcal{F}} \ell\big(Y, f(X)\big) \leq V$ a.s. for a constant $V \geq 1$. Then for all $n \geq 5$ and $\delta \leq \frac{1}{2}$, with probability at least $1 - \delta$*

$$\mathsf{P}\,\ell(\cdot, \hat{f}_{\mathbf{z}}) \leq \mathsf{P}\,\ell(\cdot, f^*) + \frac{1}{n} \max \left\{ \begin{array}{c} 8 \max\left\{V, \frac{1}{\eta^*}\right\} \left(\mathcal{C}\log(Kn) + \log\frac{2}{\delta}\right), \\ 2V\left(1080\mathcal{C}\log(2Kn) + 90\sqrt{\left(\log\frac{2}{\delta}\right)\mathcal{C}\log(2Kn)} + \log\frac{2e}{\delta}\right) \end{array} \right\} + \frac{1}{n}.$$

## 5 Characterizing convexity from the perspective of risk minimization

In the following, when we say $(\ell, \mathcal{F}, \mathsf{P})$ has a *unique minimizer* we mean that any two minimizers $f_1^*, f_2^*$ of $\mathsf{P}\,\ell(\cdot, f)$ over $\mathcal{F}$ satisfy $\ell\big(Y, f_1^*(X)\big) = \ell\big(Y, f_2^*(X)\big)$ a.s. We say the excess loss class $\{\ell(\cdot, f) - \ell(\cdot, f^*) : f \in \mathcal{F}\}$ satisfies a $(\beta, B)$-*Bernstein condition with respect to* $\mathsf{P}$ for some $B > 0$ and $0 < \beta \leq 1$ if, for all $f \in \mathcal{F}$:

$$\mathsf{P}\big(\ell(\cdot, f) - \ell(\cdot, f^*)\big)^2 \leq B\left(\mathsf{P}\big(\ell(\cdot, f) - \ell(\cdot, f^*)\big)\right)^\beta. \tag{12}$$

It already is known that the stochastic mixability condition guarantees that there is a unique minimizer [21]; this is a simple consequence of Jensen's inequality. This leaves open the question: if stochastic mixability does not hold, are there necessarily non-unique minimizers? We show that in a certain sense this is indeed the case, in bad way: the set of minimizers will be a disconnected set.

For any $\varepsilon > 0$, define $\mathcal{G}_\varepsilon$ as the class $\mathcal{G}_\varepsilon := \{f^*\} \cup \{f \in \mathcal{F} : \|f - f^*\|_{L_1(\mathsf{P})} \geq \varepsilon\}$, where in case there are multiple minimizers in $\mathcal{F}$ we arbitrarily select one of them as $f^*$. Since we assume that $\mathcal{F}$ is compact and $G_\varepsilon \setminus \{f^*\}$ is equal to $\mathcal{F}$ minus an open set homeomorphic to the unit $L_1(\mathsf{P})$ ball, $\mathcal{G}_\varepsilon \setminus \{f^*\}$ is also compact.

**Theorem 8 (Non-Unique Minimizers).** *Suppose there exists some $\varepsilon > 0$ such that $\mathcal{G}_\varepsilon$ is not stochastically mixable. Then there are minimizers $f_1^*, f_2^* \in \mathcal{F}$ of $\mathsf{P}\,\ell(\cdot, f)$ over $\mathcal{F}$ such that it is not the case that $\ell\big(Y, f_1^*(X)\big) = \ell\big(Y, f_2^*(X)\big)$ a.s.*

*Proof.* Select $\varepsilon > 0$ as in the theorem and some fixed $\eta > 0$. Since $\mathcal{G}_\varepsilon$ is not $\eta$-stochastically mixable, there exists $f_\eta \in \mathcal{G}_\varepsilon$ such that $\Lambda_{-Z_{f_\eta}}(\eta) > 0$. Note that there exists $\eta' \in (0, \eta)$ with $\Lambda_{-Z_{f_\eta}}(\eta') = 0$; if not, $\lim_{\eta \downarrow 0} \frac{\Lambda_{-Z_{f_\eta}}(\eta) - \Lambda_{-Z_{f_\eta}}(0)}{\eta} > 0 \Rightarrow \Lambda'_{-Z_{f_\eta}}(0) > 0$, so $\Lambda'_{-Z_{f_\eta}}(0) = \mathsf{E}(-Z_{f_\eta})$ implies that $\mathsf{E}\,Z_{f_\eta} < 0$, a contradiction! From Lemma 2, $\mathsf{E}\,Z_{f_\eta} \leq \frac{\cosh(\eta') - 1}{\sinh(\eta')}$; for $\eta' \geq 0$ the RHS has upper bound $\frac{\eta'}{2}$ since the derivative of $\frac{\eta'}{2} - \frac{\cosh(\eta') - 1}{\sinh(\eta')}$ is the nonnegative function $\frac{1}{2}\tanh^2(\eta'/2)$ and $\left(\frac{\eta'}{2} - \frac{\cosh(\eta') - 1}{\sinh(\eta')}\right)\big|_{\eta' = 0} = 0$. Thus, $\mathsf{E}\,Z_{f_\eta} \to 0$ as $\eta \to 0$. As $\mathcal{G}_\varepsilon \setminus \{f^*\}$ is compact, we can take a positive decreasing sequence $(\eta_j)_j$ approaching 0, corresponding to a sequence $(f_{\eta_j})_j \subset \mathcal{G}_\varepsilon \setminus \{f^*\}$ with limit point $g^* \in \mathcal{G}_\varepsilon \setminus \{f^*\}$ for which $\mathsf{E}\,Z_{g^*} = 0$, and so there is a risk minimizer in $\mathcal{G}_\varepsilon \setminus \{f^*\}$. □

**The implications of having non-unique risk minimizers.** In the case of non-unique risk minimizers, Mendelson [17] showed that for $p$-losses $(y, \hat{y}) \mapsto |y - \hat{y}|^p$ with $p \in [2, \infty)$ there is an $n$-indexed sequence of probability measures $(\mathsf{P}^{(n)})_n$ approaching the true probability measure as $n \to \infty$ such that, for each $n$, ERM learns at a slow rate under sample size $n$ when the true distribution is $\mathsf{P}^{(n)}$. This behavior is a consequence of the statistical learning problem's poor geometry: there are multiple minimizers and the set of minimizers is not even connected. Furthermore, in this case, the best known fast rate upper bounds (see [18] and [19]) have a multiplicative constant that approaches $\infty$ as the target probability measure approaches a probability measure for which there are non-unique minimizers. The reason for the poor upper bounds in this case is that the constant $B$ in the Bernstein condition explodes, and the upper bounds rely upon the Bernstein condition.

## 6 Weak stochastic mixability

For some $\kappa \in [0, 1]$, we say $(\ell, \mathcal{F}, \mathsf{P})$ is $(\kappa, \eta_0)$-weakly stochastically mixable if, for every $\varepsilon > 0$, for all $f \in \{f^*\} \cup \mathcal{F}_{\succeq \varepsilon}$, the inequality $\log \mathsf{E}\exp(-\eta_\varepsilon Z_f) \leq 0$ holds with $\eta_\varepsilon := \eta_0 \varepsilon^{1-\kappa}$. This concept was introduced by Van Erven et al. [21] without a name.

Suppose that some fixed function has excess risk $a = \varepsilon$. Then, roughly, with high probability ERM does not make a mistake provided that $a\eta_a = \frac{1}{n}$, i.e. when $\varepsilon \cdot \eta_0 \varepsilon^{1-\kappa} = \frac{1}{n}$ and hence when $\varepsilon = (\eta_0 n)^{-1/(2-\kappa)}$. Modifying the proof of the finite classes result (Theorem 5) to consider all functions in the subclass $\mathcal{F}_{\succeq \gamma_n}$ for $\gamma_n = (\eta_0 n)^{-1/(2-\kappa)}$ yields the following corollary of Theorem 5.

**Corollary 9.** *Let* $(\ell, \mathcal{F}, \mathsf{P})$ *be* $(\kappa, \eta_0)$-*weakly stochastically mixable for some* $\kappa \in [0,1]$, *where* $|\mathcal{F}| = N$, $\ell$ *is a nonnegative loss, and* $\sup_{f \in \mathcal{F}} \ell\big(Y, f(X)\big) \leq V$ *a.s. for a constant* $V$. *Then for any* $n \geq \frac{1}{\eta_0} V^{(1-\kappa)/(2-\kappa)}$, *with probability at least* $1 - \delta$

$$\mathsf{P}\,\ell(\cdot, \hat{f}_{\mathbf{z}}) \leq \mathsf{P}\,\ell(\cdot, f^*) + \frac{6\left(\log \frac{1}{\delta} + \log N\right)}{(\eta_0 n)^{1/(2-\kappa)}}.$$

It is simple to show a similar result for VC-type classes; the $\varepsilon$-net can still be taken at the resolution $\frac{1}{n}$, but we need only apply the analysis to the subclass of $\mathcal{F}$ with excess risk at least $(\eta_0 n)^{-1/(2-\kappa)}$.

# 7 Discussion

We have shown that stochastic mixability implies fast rates for VC-type classes, using a direct argument based on the Cramér-Chernoff method and sufficient control of the optimal value of a certain instance of the general moment problem. The approach is amenable to localization in that the analysis separately controls the probability of large deviations for individual elements of $\mathcal{F}$. An important open problem is to extend the results presented here for VC-type classes to results for nonparametric classes with polynomial metric entropy, and moreover, to achieve rates similar to those obtained for these classes under the Bernstein condition.

There are still some unanswered questions with regards to the connection between the Bernstein condition and stochastic mixability. Van Erven et al. [21] showed that for bounded losses the Bernstein condition implies stochastic mixability. Therefore, when starting from a Bernstein condition, Theorem 5 offers a different path to fast rates. An open problem is to settle the question of whether the Bernstein condition and stochastic mixability are equivalent. Previous results [21] suggest that the stochastic mixability does imply a Bernstein condition, but the proof was non-constructive, and it relied upon a bounded losses assumption. It is well known (and easy to see) that both stochastic mixability and the Bernstein condition hold only if there is a unique minimizer. Theorem 8 shows in a certain sense that if stochastic mixability does not hold, then there cannot be a unique minimizer. Is the same true when the Bernstein condition fails to hold? Regardless of whether stochastic mixability is equivalent to the Bernstein condition, the direct argument presented here and the connection to classical mixability, which does characterize constant regret in the simpler non-stochastic setting, motivates further study of stochastic mixability.

Finally, it would be of great interest to discard the bounded losses assumption. Ignoring the dependence of the metric entropy on the maximum possible loss, the upper bound on the loss $V$ enters the final bound through the difficulty of controlling the minimum value of $u_\eta(-1)$ when $\eta$ is large (see the proof of Theorem 3). From extensive experiments with a grid-approximation linear program, we have observed that the worst (CGF-wise) random variables for fixed negative mean and fixed optimal stochastic mixability constant are those which place very little probability mass at $-V$ and most of the probability mass at a small positive number that scales with the mean. These random variables correspond to functions that with low probability beat $f^*$ by a large (loss) margin but with high probability have slightly higher loss than $f^*$. It would be useful to understand if this exotic behavior is a real concern and, if not, find a simple, mild condition on the moments that rules it out.

**Acknowledgments**

RCW thanks Tim van Erven for the initial discussions around the Cramér-Chernoff method during his visit to Canberra in 2013 and for his gracious permission to proceed with the present paper without him as an author, and both authors thank him for the further enormously helpful spotting of a serious error in our original proof for fast rates for VC-type classes. This work was supported by the Australian Research Council (NAM and RCW) and NICTA (RCW). NICTA is funded by the Australian Government through the Department of Communications and the Australian Research Council through the ICT Centre of Excellence program.

## Footnotes

[1]See e.g. the techniques of Massart and Nédélec [16] and equation (3.17) of Koltchinskii [11].

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
