[Supplementary Material]

# A Proof of Theorem 3

*Proof of Theorem 3.* From Theorem 3 of Kemperman [9], if the moment values vector $\left(-\frac{a}{n}, 1\right)$ is in $\operatorname{int} \operatorname{conv} g([-1, 1])$, the optimal objective value of problem (6) is equal to

$$\sup\left\{d_0 - \frac{a}{n}d_1 + d_2 : d^* = (d_0, d_1, d_2) \in D^*\right\}. \tag{13}$$

From the Feasible Moments Lemma (Lemma 2 in the paper), we see that (10) corresponds to the interior point condition.

Since we assume the interior point condition is satisfied, any $d^* \in D^*$ provides a lower bound on the optimal value of (6), and hence after negation provides an upper bound on the problem with same moment constraints and the objective $\sup \mathsf{E}\, e^{(\eta/2)X}$; this is precisely what we are after.

We therefore focus on picking a good $d^* = (d_0, d_1, d_2) \in \mathbb{R}^3$. The inequality condition in (8) is now

$$-e^{(\eta/2)x} \geq d_0 + d_1 x + d_2 e^{\eta x} \quad \text{for all } x \in [-1, 1].$$

In particular, this inequality must hold at $x = 0$, yielding the constraint $-1 \geq d_0 + d_2$. We now change variables to $c_0 = -d_0$, $c_1 = -d_1/\eta$,[2] and $c_2 = -d_2$, yielding the inequality condition

$$u_\eta(x) := -e^{(\eta/2)x} + c_0 + c_2 e^{\eta x} + \eta c_1 x \geq 0.$$

Now the condition at $x = 0$ implies that $c_0 + c_2 = 1$, and so we make the replacement

$$c_0 = 1 - c_2, \tag{14}$$

in the definition of $u_\eta(x)$, yielding the inequality

$$u_\eta(x) = 1 + c_2(e^{\eta x} - 1) - e^{(\eta/2)x} + \eta c_1 x \geq 0.$$

**Constraints from the local minimum at 0**

Since $u_\eta(0) = 0$, we need $x = 0$ to be a local minimum of $u$, and so we require both conditions

(a) $u'(0) = 0$

(b) $u''(0) \geq 0$

to hold since otherwise there exists some small $\varepsilon > 0$ such that either $u_\eta(\varepsilon) < 0$ or $u_\eta(-\varepsilon) < 0$.

For (a), we compute

$$u'(x) = \eta c_2 e^{\eta x} - \frac{\eta}{2}e^{(\eta/2)x} + \eta c_1.$$

Since we require $u'(0) = 0$, we pick up the constraint

$$\eta\left(c_2 - \frac{1}{2} + c_1\right) = 0,$$

and since $\eta > 0$ by assumption, we have

$$c_1 = \frac{1}{2} - c_2. \tag{15}$$

Thus, we can eliminate $c_1$ from $u_\eta(x)$:

$$u_\eta(x) = 1 + c_2(e^{\eta x} - 1) - e^{(\eta/2)x} + \eta\left(\frac{1}{2} - c_2\right)x \geq 0.$$

For (b), it is sufficient to have $u''(0) > 0$. Observe that

$$u''(x) = \eta^2 c_2 e^{\eta x} - \frac{\eta^2}{4}e^{(\eta/2)x},$$

so that $u''(0) = \eta^2\left(c_2 - \frac{1}{4}\right)$, and hence for

$$c_2 > \frac{1}{4} \tag{16}$$

we have $u''(0) > 0$.

Thus far, we have picked up the constraints (14), (15), and (16).

**The other minima of $u_\eta(x)$**

Now, observe that $u'(x)$ has at most two roots, because with the substitution $y = e^{(\eta/2)x}$, we have

$$u'(x) = \eta c_2 y^2 - \frac{\eta}{2} y + \eta \left( \frac{1}{2} - c_2 \right),$$

which is a quadratic equation in $y$ with two roots:

$$y = \left\{ 1, \frac{1 - 2c_2}{2c_2} \right\} \quad \Rightarrow \quad x = \left\{ 0, \frac{2}{\eta} \log \frac{1 - 2c_2}{2c_2} \right\}.$$

Now, since we take $c_2 > \frac{1}{4}$ and since the second root is negative, we know that $u$ is increasing on $[0, 1]$ (and we already knew that $u_\eta(0) = 0$). It remains to find conditions on $c_2$ such that $u_\eta(-1) \geq 0$ because that implies that $u_\eta(x) \geq 0$ for all $x \in [-1, 0]$. We consider the case $\eta \leq 1$ and $\eta > 1$ separately.

In either case, we need to check the nonnegativity of

$$u_\eta(-1) = 1 + c_2(e^{-\eta} - 1) - e^{-(\eta/2)} - \eta \left( \frac{1}{2} - c_2 \right)$$

$$= \left( 1 - \frac{\eta}{2} \right) - e^{-(\eta/2)} + c_2 \left( e^{-\eta} - (1 - \eta) \right).$$

**Case $\eta \leq 1$:** We observe that $u_\eta(-1) = 0$ when $\eta = 0$. Now, we will see what constraints on $c_2$ guarantee that $\frac{d}{d\eta} u_\eta(-1) \geq 0$ for $\eta \in [0, 1]$. We *want*

$$\frac{d}{d\eta} u_\eta(-1) = -c_2 e^{-\eta} + \frac{1}{2} e^{-\eta/2} - \frac{1}{2} + c_2 \geq 0$$

which is equivalent to the condition

$$c_2 \geq \frac{1}{2} \left( \frac{1 - e^{-\eta/2}}{1 - e^{-\eta}} \right).$$

The RHS is increasing in $\eta$, and so we need only consider $\eta = 1$, yielding the bound

$$c_2 \geq \frac{1}{2} \frac{e - \sqrt{e}}{e - 1} = 0.3112\ldots,$$

and so if $c_2 \geq 0.32$, then $u_\eta(-1) \geq 0$ as desired.

**Case $\eta > 1$:** Let $c_2 = \frac{1}{2} - \frac{\alpha}{\eta}$ for some $\alpha \geq 0$. With this substitution, we have

$$u_\eta(-1) = 1 + c_2(e^{-\eta} - 1) - e^{-(\eta/2)} - \eta \left( \frac{1}{2} - c_2 \right)$$

$$= 1 + \left( \frac{1}{2} - \frac{\alpha}{\eta} \right)(e^{-\eta} - 1) - e^{-(\eta/2)} - \alpha$$

$$= \left( \frac{1 + e^{-\eta}}{2} - e^{-\eta/2} \right) + \alpha \left( -1 + \frac{1}{\eta} \left( 1 - e^{-\eta} \right) \right)$$

Since we want the above to be nonnegative for all $\eta > 1$, we arrive at the condition

$$\alpha \leq \inf_{\eta \geq 1} \left\{ \frac{\frac{1 + e^{-\eta}}{2} - e^{-\eta/2}}{1 - \frac{1}{\eta} \left( 1 - e^{-\eta} \right)} \right\} \tag{17}$$

Plotting suggests that the minimum is attained at $\eta = 1$, with the value $\frac{1}{2}(\sqrt{e} - 1)^2$. We will fix $\alpha$ to this value and verify that

$$\left( \frac{1 + e^{-\eta}}{2} - e^{-\eta/2} \right) + \left( \frac{1}{2}(\sqrt{e} - 1)^2 \right) \left( -1 + \frac{1}{\eta} \left( 1 - e^{-\eta} \right) \right) \geq 0. \tag{18}$$

This is true with equality at $\eta = 0$. The derivative of the LHS with respect to $\eta$ is

$$\frac{1}{2}e^{-\eta}\left(e^{\eta/2} - 1 - \frac{(\sqrt{e}-1)^2(e^\eta - \eta - 1)}{\eta^2}\right).$$

The derivative is positive at $\eta = 1$, so 0 is a candidate minimum. Eventually, $\frac{(\sqrt{e}-1)^2(e^\eta - \eta - 1)}{\eta^2}$ grows more quickly than $e^{\eta/2} - 1$ and surpasses the latter in value. The derivative is therefore negative for all sufficiently large $\eta$, and so we need only take the minimum of the LHS of (18) evaluated at $\eta = 1$ and the limiting value as $\eta \to \infty$. We have

$$\lim_{\eta\to\infty}\left(\frac{1+e^{-\eta}}{2} - e^{-\eta/2}\right) + \left(\frac{1}{2}(\sqrt{e}-1)^2\right)\left(-1 + \frac{1}{\eta}\left(1 - e^{-\eta}\right)\right) = \sqrt{e} - \frac{e}{2} \geq 0$$

Hence, (18) indeed holds for $\alpha = \frac{1}{2}(\sqrt{e}-1)^2$. We conclude that $u_\eta(-1) \geq 0$ when $\alpha \leq \frac{1}{2}(\sqrt{e}-1)^2$.

**Putting it all together**

In the regime $\eta \leq 1$, we have the constraints $c_2 > \frac{1}{4}$ and $c_2 \geq \frac{1}{2}\frac{e-\sqrt{e}}{e-1}$ (which exceeds $\frac{1}{4}$), so we can choose

$$c_1 = \frac{1}{2} - c_2 = \frac{1}{2} - \frac{1}{2}\frac{e - \sqrt{e}}{e - 1} = \frac{1}{2}\frac{\sqrt{e}-1}{e-1} = 0.1877\ldots.$$

In the regime $\eta > 1$, we have the constraints $c_2 > \frac{1}{4}$ and $\alpha \leq \frac{1}{2}(\sqrt{e}-1)^2 \Rightarrow c_2 \geq \frac{1}{2} - \frac{1}{2\eta}(\sqrt{e}-1)^2$ (which always exceeds $\frac{1}{4}$ for $\eta \geq 1$), so we can choose

$$c_1 = \frac{1}{2} - c_2 = \frac{\alpha}{\eta} \leq \frac{(\sqrt{e}-1)^2}{2\eta} = \frac{0.2104\ldots}{\eta}.$$

The result follows by observing that in the case of $\eta \leq 1$, the supremum in (13) is lower bounded by $-1 + \frac{0.18a\eta}{n}$, and hence the optimal objective value of (6) is lower bounded by the same quantity. Therefore, the problem with the same constraints and the objective $\sup_{\mu\in[-1,1]}\mathsf{E}\,e^{(\eta/2)X}$ has its optimal objective value upper bounded by $1 - \frac{0.18a\eta}{n}$. Repeat the same argument for the case of $\eta > 1$. $\qquad\square$

*Proof of Lemma 4.* Let $X$ be a random variable taking values in $[-V, V]$ with mean $-\frac{a}{n}$ and $\mathsf{E}\,e^{\eta X} = 1$, and let $Y$ be a random variable taking values in $[-1, 1]$ with mean $-\frac{a/V}{n}$ and $\mathsf{E}\,e^{(V\eta)Y} = 1$. Consider a random variable $\tilde{X}$ that is a $\frac{1}{V}$-scaled independent copy of $X$; observe that $\mathsf{E}\,\tilde{X} = -\frac{a/V}{n}$ and $\mathsf{E}\,e^{(V\eta)\tilde{X}} = 1$. Let the maximal possible value of $\mathsf{E}\,e^{(\eta/2)X}$ be $b_X$, and let the maximal possible value of $\mathsf{E}\,e^{(V\eta/2)Y}$ be $b_Y$. We claim that $b_X = b_Y$. Let $X$ be a random variable with a distribution that maximizes $\mathsf{E}\,e^{(\eta/2)X}$ subject to the previously stated constraints on $X$. Since $\tilde{X}$ satisfies $\mathsf{E}\,e^{(V\eta/2)\tilde{X}} = b_X$, setting $Y = \tilde{X}$ shows that in fact $b_Y \geq b_X$. A symmetric argument (starting with $Y$ and passing to some $\tilde{Y} = VY$) implies that $b_X \geq b_Y$. $\qquad\square$

## B  Hyper-concentrated excess losses

**Lemma 10.** *Let $Z$ be a random variable with probability measure $P$ supported on $[-V, V]$. Suppose that $\lim_{\eta\to\infty}\mathsf{E}\exp(-\eta Z) < 1$ and $\mathsf{E}\,Z = \mu > 0$. Then there is a suitable modification of $Z'$ for which $Z' \leq Z$ with probability 1, the mean of $Z'$ is arbitrarily close to $\mu$, and $\mathsf{E}\exp(-\eta Z') = 1$ for arbitrarily large $\eta$.*

*Proof.* First, observe that $Z \geq 0$ a.s. If not, then there must be some finite $\eta > 0$ for which $\mathsf{E}\exp(-\eta Z) = 1$. Now, consider a random variable $Z'$ with probability measure $Q_\epsilon$, a modification of $Z$ (with probability measure $P$) constructed in the following way. Define $A := [\mu, V]$ and $A^- := [-V, -\mu]$. Then for any $\epsilon > 0$ we define $Q_\epsilon$ as

$$dQ_\epsilon(z) = \begin{cases} (1-\epsilon)dP(z) & \text{if } z \in A \\ \epsilon dP(-z) & \text{if } z \in A^- \\ dP(z) & \text{otherwise.} \end{cases}$$

Additionally, we couple $P$ and $Q_\varepsilon$ such that the couple $(Z, Z')$ is a coupling of $(P, Q_\epsilon)$ satisfying

$$\mathsf{E}_{(Z,Z')\sim(P,Q_\epsilon)}\,\mathbf{1}_{\{Z\neq Z'\}} = \min_{(P',Q'_\epsilon)} \mathsf{E}_{(Z,Z')\sim(P',Q'_\epsilon)}\,\mathbf{1}_{Z\neq Z'},$$

where the $\min$ is over all couplings of $P$ and $Q_\varepsilon$. This coupling ensures that $Z' \leq Z$ with probability 1; i.e. $Z'$ is dominated by $Z$.

Now,

$$
\begin{aligned}
\mathsf{E}\exp(-\eta Z') &= \int_{-V}^{V} e^{-\eta z}\,dQ_\epsilon(z) \\
&= \int_{A^-} e^{-\eta z}\,dQ_\epsilon(z) + \int_A e^{-\eta z}\,dQ_\epsilon(z) + \int_{[0,V]\setminus A} e^{-\eta z}\,dQ_\epsilon(z) \\
&= \epsilon\int_{A^-} e^{-\eta z}\,dP(-z) + (1-\epsilon)\int_A e^{-\eta z}\,dP(z) + \int_{[0,V]\setminus A} e^{-\eta z}\,dP(z) \\
&= \epsilon\int_A e^{\eta z}\,dP(z) + (1-\epsilon)\int_A e^{-\eta z}\,dP(z) + \int_{[0,V]\setminus A} e^{-\eta z}\,dP(z) \\
&\geq \epsilon e^{\mu\eta}P(A) + (1-\epsilon)\int_A e^{-\eta z}\,dP(z) + \int_{[0,V]\setminus A} e^{-\eta z}\,dP(z). \qquad (19)
\end{aligned}
$$

Now, on the one hand, for any $\eta > 0$, the sum of the two right-most terms in (19) is strictly less than 1 by assumption. On the other hand, $\eta \to \epsilon P(A)e^{\mu\eta}$ is exponentially increasing since $\epsilon > 0$ and $\mu > 0$ (and hence $P(A) > 0$ as well) by assumption; thus, the first term in (19) can be made arbitrarily large for large enough by increasing $\eta$. Consequently, we can choose $\epsilon > 0$ as small as desired and then choose $\eta < \infty$ as large as desired such that the mean of $Z'$ is arbitrarily close to $\mu$ and $\mathsf{E}\exp(-\eta Z') = 1$ respectively. $\qquad\square$

## C  Proof of VC-type results

### C.1  Proof of Theorem 6

The localization result (Theorem 6) is a simple consequence of the following theorem.

**Theorem 11 (Local Analysis).** *Let $\mathcal{F} \subset \mathbb{R}^\mathcal{X}$ be a separable function class for which:*

- *the constant zero function is an element of $\mathcal{F}$ ;*

- *every function $f \in \mathcal{F}$ satisfies $0 \leq f \leq 1$ ;*

- $\sup_{f\in\mathcal{F}} \|f\|_{L_2(\mathsf{P})} \leq \varepsilon := \frac{1}{n}$ .

*Further assume that for some $\mathcal{C} \geq 1$, for a constant $K \geq 1$, for each $u \in (0, K]$ the $L_2(\mathsf{P})$ covering numbers of $\mathcal{F}$ are bounded as*

$$\mathcal{N}(u, \mathcal{F}, L_2(\mathsf{P})) \leq \left(\frac{K}{u}\right)^\mathcal{C}.$$

*Then provided that $n \geq 4$ and $y > 0$, with probability at least $1 - e^{-y}$*

$$\sup_{f\in\mathcal{F}} \mathsf{P}_n f \leq \frac{1}{n}\left(990\mathcal{C}\log(2Kn) + \sqrt{2y(1 + 3960\mathcal{C}\log(2Kn))} + \frac{2y}{3} + 1\right).$$

REMARKS

(i) The class $\mathcal{F}$ is contained in an $L_2(\mathsf{P})$-ball of radius $\varepsilon$, and if interpreted as a loss class it is assumed that the losses are bounded.

(ii) Suppose the function class $\mathcal{F}$ is constructed by selecting from a larger class an $\varepsilon$-ball in the $L_2(\mathsf{P})$ pseudometric around some function $f_0$ from the same larger class and taking for each function the absolute difference with $f_0$. Then the zero function trivially is in $\mathcal{F}$ since $|f_0 - f_0|$ is in the class. In this setup, the theorem states that with high probability there is no function in the class whose empirical risk will be "much" smaller/larger than the empirical risk of $f_0$.

*Proof of Theorem 11.* For the proof, we introduce the random variables $Z = \sup_{f \in \mathcal{F}} \mathsf{P}_n f$ and $\bar{Z} = \sup_{f \in \mathcal{F}} (\mathsf{P}_n - \mathsf{P}) f$. The proof is in three steps.

**Step 1: Centering approximation**

It is easy to see that $Z \leq \bar{Z} + \varepsilon$, since

$$
\begin{aligned}
Z = \sup_{f \in \mathcal{F}} \mathsf{P}_n f = \sup_{f \in \mathcal{F}} (\mathsf{P}_n - \mathsf{P}) f + \mathsf{P} f \\
\leq \sup_{f \in \mathcal{F}} (\mathsf{P}_n - \mathsf{P}) f + \sup_{f \in \mathcal{F}} \mathsf{P} f \\
\leq \bar{Z} + \sup_{f \in \mathcal{F}} \|f_0 - f\|_{L_1(\mathsf{P})} \\
\leq \bar{Z} + \sup_{f \in \mathcal{F}} \|f_0 - f\|_{L_2(\mathsf{P})} \\
\leq \bar{Z} + \varepsilon,
\end{aligned}
$$

where the penultimate inequality follows from Jensen's inequality.

**Step 2: Concentration of $\bar{Z}$ arounds its expectation**

We will apply Bousquet's version of Talagrand's inequality, appearing as equation (18) of [16] and reproduced below for convenience:

> If $\mathcal{G}$ is a countable family of measurable functions such that, for some positive constants $v$ and $b$, one has, for every $g \in \mathcal{G}$, $\mathsf{P} g^2 \leq v$ and $\|g\|_\infty \leq b$, then, for every positive $y$, the following inequality holds for $W = \sup_{g \in \mathcal{G}} (\mathsf{P}_n - \mathsf{P}) g$:
>
> $$ \Pr\left\{ W - \mathsf{E}\, W \geq \sqrt{\frac{2(v + 4b\,\mathsf{E}\, W)y}{n}} + \frac{2by}{3n} \right\} \leq e^{-y}. $$

We take $\mathcal{G}$ to be $\mathcal{F}$ itself; since $\mathcal{F}$ is separable and hence admits a countable dense subset, the countability assumption in Talagrand's inequality is not an issue. Observe that for every $f \in \mathcal{F}$ we have

- $\|f\|_\infty \leq 1$ (from the range constraints on $f$)

- $\mathsf{P} f^2 \leq \|f\|_{L_2(\mathsf{P})} \leq \varepsilon$ (by the small $L_2(\mathsf{P})$-ball assumption on $\mathcal{F}$).

Thus, taking $b = 1$ and $v = \varepsilon$, we have

$$ \Pr\left\{ \bar{Z} - \mathsf{E}\, \bar{Z} \geq \sqrt{\frac{2(\varepsilon + 4\,\mathsf{E}\, \bar{Z})y}{n}} + \frac{2y}{3n} \right\} \leq e^{-y}. \tag{20} $$

It remains to bound $\mathsf{E}\, \bar{Z}$. If it can be shown to be $\tilde{O}(\frac{1}{n})$ then we will have the desired result after taking $\varepsilon = O(\frac{1}{n})$.

**Step 3: Controlling the size of $\mathsf{E}\, \bar{Z}$**

Controlling $\mathsf{E}\, \bar{Z}$ can be done through chaining after passing to a symmetrized empirical process. This control is shown in Lemma 12, stated after the current proof, yielding the result

$$ \mathsf{E}\, \bar{Z} \leq \frac{990\mathcal{C} \log(2Kn)}{n}. \tag{21} $$

**Putting it all together**

The desired result follows by plugging (21) into the concentration result (20), incorporating the $\varepsilon$ approximation term from Step 1, and setting $\varepsilon = \frac{1}{n}$. $\qquad\square$

**Lemma 12.** *Take the same conditions as Theorem 11 (Local Analysis Theorem), but instead allow that all $f \in \mathcal{F}$ need only satisfy $0 \le f \le V$ for some $V \ge 1$. Then provided that $n \ge 4$,*

$$\mathsf{E} \sup_{f \in \mathcal{F}} (\mathsf{P}_n - \mathsf{P})f \le \frac{990 \mathcal{C} V \log(2Kn)}{n}$$

*Proof.* To avoid measurability issues, we operate under the assumption that $\mathcal{F}$ is countable and in the final step of the proof apply an approximation argument.

Let $\epsilon_1, \ldots, \epsilon_n$ be iid Rademacher random variables. We write $\mathsf{E}_\epsilon$ for the expected value with respect to the random variables $\epsilon_1, \ldots, \epsilon_n$. That is, if $A$ is a random variable depending only on $\epsilon_1, \ldots, \epsilon_n, X_1, \ldots, X_n$, then $\mathsf{E}_\epsilon A = \mathsf{E}[A \mid X_1, \ldots, X_n]$. Also, let $\mathcal{F}_{|\mathbf{x}}$ be the coordinate projection of $\mathcal{F}$ onto the sample $\mathbf{X} = (X_1, \ldots, X_n)$:

$$\mathcal{F}_{|\mathbf{x}} := \big\{ (f(X_1), \ldots, f(X_n)) : f \in \mathcal{F} \big\}.$$

Finally, for a set $\mathcal{G} \subset \mathbb{R}^n$ let $D(\mathcal{G})$ be half of the $\ell_2$-radius of $\mathcal{G}$, defined as

$$D(\mathcal{G}) := \frac{1}{2} \sup_{g \in \mathcal{G}} \|g\|_2;$$

it makes sense to refer to this as a (half) radius since we will consider $D(\mathcal{F})$ and the zero function is in $\mathcal{F}$. Our life will be made easier if we use the lower bounded quantity $D(\mathcal{G}) \vee \sigma$, for some deterministic $\sigma \le 1$ to be chosen later.

The first step is symmetrization. The second step is based on a chaining argument, the result of which is Corollary 13.2 of Boucheron et al. [5], restated here[3] we state the result in terms of in a specialization to Rademacher processes for convenience:

> Let $(\mathcal{T}, d)$ be a finite pseudometric space and $(X_t)_{t \in \mathcal{T}}$ be a collection of sub-Gaussian random variables. Then for any $t_0 \in \mathcal{T}$,
> $$\mathsf{E} \sup_{t \in \mathcal{T}} X_t - X_{t_0} \le 12 \int_0^{\delta/2} \sqrt{\log \mathcal{N}(u, \mathcal{T}, d)} du,$$
> where $\delta = \sup_{t \in \mathcal{T}} d(t, t_0)$.

By pushing the cardinality of $\mathcal{T}$ to infinity, the above result also applies to countable classes. As noted by Boucheron et al. [5] in the paragraph concluding the statement of their Corollary 13.2, this result applies to Rademacher processes. In our case, $t_0$ will correspond to the zero function element of $\mathcal{F}$.

Define $\bar{\mathsf{P}}_n := \mathsf{P}_n - \mathsf{P}$. Now, from symmetrization and the above chaining-based result applied to the resulting Rademacher process, we have

$$n \, \mathsf{E} \sup_{f \in \mathcal{F}} \bar{\mathsf{P}}_n f \le 2 \, \mathsf{E} \left( \mathsf{E}_\epsilon \sup_{f \in \mathcal{F}} \sum_{j=1}^n \epsilon_j f(X_j) \right)$$

$$\le 24 \, \mathsf{E} \int_0^{D(\mathcal{F}_{|\mathbf{x}}) \vee \sigma} \sqrt{\log \mathcal{N}(u, \mathcal{F}_{|\mathbf{x}}, \| \cdot \|_2)} du,$$

which (since if $f_{|X}$ is the obvious coordinate projection of $f \in \mathcal{F}$, then $\|f_{|X}\|_2 = \sqrt{n} \|f\|_{L_2(\mathsf{P}_n)}$) is at most

$$24 \, \mathsf{E} \int_0^{D(\mathcal{F}_{|\mathbf{x}}) \vee \sigma} \sqrt{\mathcal{H}_2 \left( \frac{u}{\sqrt{n}}, \mathcal{F}_{|\mathbf{x}} \right)} du$$

$$\le 24 \sqrt{\mathcal{C}} \, \mathsf{E} \int_0^{D(\mathcal{F}_{|\mathbf{x}}) \vee \sigma} \sqrt{\log \frac{K\sqrt{n}}{u}} du,$$

where $\mathcal{H}_2(u, \mathcal{T}) := \sup_Q \log \mathcal{N}(u, \mathcal{T}, L_2(Q))$ is the *universal metric entropy* of $\mathcal{T}$, and in the above display we have $\mathcal{H}_2\left(\frac{u}{\sqrt{n}}, \mathcal{F}_{|\mathbf{x}}\right)$ rather than $\mathcal{H}_2(u, \mathcal{F}_{|\mathbf{x}})$ because we work with the $L_2(\mathsf{P}_n)$-norm scaled by $\sqrt{n}$.

Making the substitution $t = u/(D(\mathcal{F}_{|\mathbf{x}}) \vee \sigma)$, the above is equal to

$$24\sqrt{\mathcal{C}}\, \mathsf{E}\left((D(\mathcal{F}_{|\mathbf{x}}) \vee \sigma) \int_0^1 \sqrt{\log \frac{K\sqrt{n}}{t(D(\mathcal{F}_{|\mathbf{x}}) \vee \sigma)}} dt\right)$$

$$\leq 24\sqrt{\mathcal{C}}\, \mathsf{E}\left((D(\mathcal{F}_{|\mathbf{x}}) \vee \sigma)\left(\sqrt{\log \frac{K\sqrt{n}}{D(\mathcal{F}_{|\mathbf{x}}) \vee \sigma}} + \frac{\sqrt{\pi}}{2}\right)\right)$$

$$\leq 24\sqrt{\mathcal{C}}\left(\sqrt{\log \frac{K\sqrt{n}}{\sigma}} + \frac{\sqrt{\pi}}{2}\right)(\sigma + \mathsf{E}\, D(\mathcal{F}_{|\mathbf{x}})).$$

Now, we focus on $\mathsf{E}\, D(\mathcal{F}_{|\mathbf{x}})$. Observe that

$$\mathsf{E}\, D(\mathcal{F}_{|\mathbf{x}}) = \frac{\sqrt{n}}{2} \mathsf{E} \sup_{f \in \mathcal{F}} \sqrt{\mathsf{P}_n f^2}$$

$$= \frac{\sqrt{n}}{2} \mathsf{E} \sup_{f \in \mathcal{F}} \sqrt{\bar{\mathsf{P}}_n f^2 + \mathsf{P}\, f^2}$$

$$\leq \frac{\sqrt{n}}{2}\left[\mathsf{E} \sup_{f \in \mathcal{F}} \sqrt{\bar{\mathsf{P}}_n f^2} + \sup_{f \in \mathcal{F}} \sqrt{\mathsf{P}\, f^2}\right]$$

$$\leq \frac{\sqrt{n}}{2}\left[\sqrt{\mathsf{E} \sup_{f \in \mathcal{F}} \bar{\mathsf{P}}_n f^2} + \varepsilon\right].$$

where the first part of the last step follows from Jensen's inequality and the second part follows from the small $L_2(\mathsf{P})$-ball assumption on $\mathcal{F}$. The above is at most

$$\frac{\sqrt{Vn}}{2}\left[\sqrt{\mathsf{E} \sup_{f \in \mathcal{F}} \bar{\mathsf{P}}_n f} + \varepsilon\right].$$

Thus, putting everything together and making the replacement $\varepsilon = \frac{1}{n}$, we have

$$\mathsf{E} \sup_{f \in \mathcal{F}} \bar{\mathsf{P}}_n f \leq \frac{24\sqrt{\mathcal{C}}}{n}\left(\sqrt{\log \frac{K\sqrt{n}}{\sigma}} + \frac{\sqrt{\pi}}{2}\right)\left(\sigma + \frac{\sqrt{Vn}}{2}\left[\sqrt{\mathsf{E} \sup_{f \in \mathcal{F}} \bar{\mathsf{P}}_n f} + \frac{1}{n}\right]\right).$$

Finding the minimal value of $\mathsf{E} \sup_{f \in \mathcal{F}} \bar{\mathsf{P}}_n$ just amounts to solving a quadratic equation, yielding the solution set

$$\sqrt{\mathsf{E} \sup_{f \in \mathcal{F}} \bar{\mathsf{P}}_n f} \leq \frac{\psi \sqrt{Vn}}{2} + \sqrt{\psi\left(\sigma + \frac{\sqrt{V/n}}{2}\right)}$$

for $\psi = \frac{24\sqrt{\mathcal{C}}}{n}\left(\sqrt{\log \frac{K\sqrt{n}}{\sigma}} + \frac{\sqrt{\pi}}{2}\right)$.

Making the replacement $\sigma = \frac{1}{n}$, squaring, and some coarse bounding yields

$$\mathsf{E} \sup_{f \in \mathcal{F}} \bar{\mathsf{P}}_n f \leq \frac{990 \mathcal{C} V \log(Kn)}{n}$$

for $n \geq 4$, $V \geq 1$, and $\mathcal{C} \geq 1$.

We now present the approximation argument to handle separable $\mathcal{F}$. Since $\mathcal{F}$ is separable, it suffices to consider a countable dense subset $\mathcal{F}' \subset \mathcal{F}$; however, a little more work is required as the covering

numbers of $\mathcal{F}'$ may differ slightly from the covering numbers of $\mathcal{F}$. We now control the covering numbers of $\mathcal{F}'$ in terms of the covering numbers of $\mathcal{F}$. Observe that if there is an $\varepsilon$-net of $\mathcal{F}$ of cardinality $N$, then there is a $(2\varepsilon)$-net of some $\mathcal{F}' \subset \mathcal{F}$ of cardinality $N$. Hence, if there is an optimal $\varepsilon$-net of $\mathcal{F}$ of cardinality $N$, then an optimal $(2\varepsilon)$-net of $\mathcal{F}'$ has cardinality at most $N$. That is, for any probability measure $Q$ on $\mathcal{X}$, for any $u > 0$ we have $\mathcal{N}(2u, \mathcal{F}', L_2(Q)) \leq \mathcal{N}(u, \mathcal{F}, L_2(Q))$. Thus, the result for separable $\mathcal{F}$ holds by replacing the constant $K$ with $2K$. □

We now prove the localization result.

*Proof of Theorem 6.* First, so that we can just handle the case of functions with range $[0, 1]$, we (crudely) apply our analysis to the function class after scaling all functions by the factor $\frac{1}{V}$, and scale the approximation term in the last step by the factor $V$.[4]

For any $f_0 \in \mathcal{F}_\varepsilon$, observe that $\pi^{-1}(f_0)$ is the set of those functions that are covered by $f_0$ in the $L_2(\mathsf{P})$-norm. We apply the Local Analysis Theorem (Theorem 11) to each element of the set of localized *absolute difference* function classes

$$\{\mathcal{G}_{f_0} : f_0 \in \mathcal{F}_\varepsilon\},$$

for $\mathcal{G}_{f_0} := \{|f_0 - f| : f \in \pi^{-1}(f_0)\}$. Consider an arbitrary $f_0 \in \mathcal{F}_\varepsilon$ and its corresponding class $\mathcal{G}_{f_0}$. Since $\tilde{\mathcal{G}}_{f_0} := \{f_0 - f : f \in \pi^{-1}(f_0)\}$ is isomorphic to a subset of $\mathcal{F}$, and since any $\varepsilon$-net for $\tilde{G}_{f_0}$ trivially gives rise to an $\varepsilon$-net for $\mathcal{G}_{f_0}$ by taking the absolute value of each function from the original $\varepsilon$-net, it follows the $L_2(\mathsf{P})$ covering numbers of $\mathcal{G}_{f_0}$ are bounded just as in (11).

Taking the union bound over $\mathcal{F}_\varepsilon$ with Theorem 11 implies that with probability at least $1 - \delta$

$$\max_{f_0 \in \mathcal{F}_\varepsilon} \sup_{f \in \pi^{-1}(f_0)} \mathsf{P}_n |f_0 - f|$$

$$\leq \frac{1}{n}\left(990\mathcal{C}\log(2Kn) + \sqrt{2\left(\log\frac{1}{\delta} + \mathcal{C}\log(Kn)\right)(1 + 3960\mathcal{C}\log(2Kn))}\right)$$

$$+ \frac{1}{n}\left(\frac{2\left(\log\frac{1}{\delta} + \mathcal{C}\log(Kn)\right)}{3} + 1\right).$$

Ignoring the $\frac{1}{n}$ factor, the RHS is at most

$$990\mathcal{C}\log(2Kn) + \sqrt{2\left(\log\frac{1}{\delta} + \mathcal{C}\log(Kn)\right)(1 + 3960\mathcal{C}\log(2Kn))} + \log\frac{e}{\delta} + \mathcal{C}\log(Kn),$$

which is at most

$$991\mathcal{C}\log(2Kn)$$

$$+ \sqrt{2\log\frac{1}{\delta} + 2\mathcal{C}\log(Kn) + 7920\left(\log\frac{1}{\delta}\right)\mathcal{C}\log(2Kn) + 7920(\mathcal{C}\log(2Kn))^2} + \log\frac{e}{\delta}$$

$$\leq 1080\mathcal{C}\log(2Kn) + \sqrt{2\log\frac{1}{\delta} + 2\mathcal{C}\log(Kn) + 7920\left(\log\frac{1}{\delta}\right)\mathcal{C}\log(2Kn)} + \log\frac{e}{\delta}$$

$$\leq 1080\mathcal{C}\log(2Kn) + 90\sqrt{\left(\log\frac{1}{\delta}\right)\mathcal{C}\log(2Kn)} + \log\frac{e}{\delta},$$

where the last inequality holds provided that $\delta$ is not too large; it suffices to assume $\delta \leq \frac{1}{2}$. □

Finally, we prove the fast rates exact oracle inequality for VC-type classes.

## C.2 Proof of Theorem 7

*Proof of Theorem 7.* For convenience, we begin by abusing notation and redefining $\mathcal{F}$ as $\mathcal{F} := \ell \circ \mathcal{F}$; the abuse includes $f^*$ being redefined as $\ell(\cdot, f^*)$. With these abuses, for any $f \in \mathcal{F}$ we redefine $Z_f$ as $Z_f := f - f^*$.

Next, we introduce a few subclasses that we will use. Recall that for any $\gamma_n > 0$, $\mathcal{F}_{\succeq \gamma_n}$ is the subclass of $\mathcal{F}$ for which the excess risk is at least $\gamma_n$. Also, for any $\gamma_n > 0$, let $\mathcal{F}_{\succeq \gamma_n, \varepsilon}$ be a proper cover of $\mathcal{F}_{\succeq \gamma_n}$ with respect to the $L_2(\mathsf{P}_n)$ norm, with $\varepsilon = \frac{1}{n}$. For each $\eta > 0$ and $\mathcal{F}_{\succeq \gamma_n, \varepsilon}$, let $\mathcal{F}^{(\eta)}_{\succeq \gamma_n, \varepsilon} \subset \mathcal{F}_{\succeq \gamma_n, \varepsilon}$ correspond to those functions for which $\eta$ is the largest constant such that $\mathsf{E} \exp(-\eta Z_f) = 1$. After making the same implicit change to $\mathcal{F}_{\succeq \gamma_n, \varepsilon}$ for "hyper-concentrated" excess loss random variables (i.e. those $Z_f$ for which $\lim_{\eta \to \infty} \mathsf{E} \exp(-\eta Z_f) < 1$) as was made to $\mathcal{F}_{\succeq \gamma_n}$ in the proof of Theorem 5, we have $\mathcal{F}_{\succeq \gamma_n, \varepsilon} = \bigcup_{\eta \in [\eta^*, \infty)} \mathcal{F}^{(\eta)}_{\succeq \gamma_n, \varepsilon}$.

Let $\gamma_n = \frac{a}{n}$ for some constant $a > 1$ to be fixed later. Consider an arbitrary $\eta \in [\eta^*, \infty)$ for which $|\mathcal{F}^{(\eta)}_{\succeq \gamma_n, \varepsilon}| > 0$, and recall that all functions $f$ in this class satisfy $\mathsf{E} Z_f \geq \frac{a}{n}$. Individually for each such function $f$, we will apply the Cramér-Chernoff Theorem (Theorem 1) as follows. From the Bounded Losses Lemma (Lemma 4), we have $\Lambda_{-Z_f}(\eta/2) = \Lambda_{-\frac{1}{V} Z_f}(V\eta/2)$. From the Stochastic Mixability Concentration Theorem (Theorem 3), the latter is at most

$$-\frac{0.18(V\eta \wedge 1)(a/V)}{n} = -\frac{0.18\eta a}{(V\eta \vee 1)n}.$$

Hence, the Cramér-Chernoff Theorem (Theorem 1) with $t = \frac{a}{2n}$ and the $\eta$ from that Theorem taken to be $\eta/2$ implies that the probability of the event $\mathsf{P}_n f \leq \mathsf{P}_n f^* + \frac{a}{2n}$ is at most

$$\exp\left(-0.18 \frac{\eta}{V\eta \vee 1} a + \frac{a\eta}{4n}\right) = \exp\left(-\eta a \left(\frac{0.18}{V\eta \vee 1} - \frac{1}{4n}\right)\right).$$

Applying the union bound over all of $\mathcal{F}_{\succeq \gamma_n, \varepsilon}$, we conclude that

$$\Pr\left\{\exists f \in \mathcal{F}_{\succeq \gamma_n, \varepsilon} : \mathsf{P}_n f \leq \mathsf{P}_n f^* + \frac{a}{2n}\right\} \leq \left(\frac{K}{\varepsilon}\right)^{\mathcal{C}} \exp\left(-\eta^* a \left(\frac{0.18}{V\eta^* \vee 1} - \frac{1}{4n}\right)\right).$$

Now, observe that if we consider some fixed failure probability $\frac{\delta}{2}$ and invert to obtain the corresponding $a$, we have

$$\begin{aligned}
a = \frac{\mathcal{C} \log \frac{K}{\varepsilon} + \log \frac{2}{\delta}}{\eta^* \left(\frac{0.18}{V\eta^* \vee 1} - \frac{1}{4n}\right)} &= \frac{\mathcal{C} \log \frac{K}{\varepsilon} + \log \frac{2}{\delta}}{\eta^* \left(\frac{0.18 - (V\eta^* \vee 1)/(4n)}{V\eta^* \vee 1}\right)} \\
&\leq \frac{(V\eta^* \vee 1) \left(\mathcal{C} \log \frac{K}{\varepsilon} + \log \frac{2}{\delta}\right)}{\eta^* \left(0.18 - \frac{1}{4n}\right)} \\
&\leq 8 \left(V \vee \left(\frac{1}{\eta^*}\right)\right) \left(\mathcal{C} \log \frac{K}{\varepsilon} + \log \frac{2}{\delta}\right) =: \lambda, \quad (22)
\end{aligned}$$

for $\gamma_n^{(1)} := \frac{\lambda}{n}$, where the last inequality holds since $n \geq 5$. Note that by instead setting $\frac{a}{n} = \gamma_n^{(1)}$ (defined in (22)) the failure probability can only decrease. Thus, for any $\gamma_n \geq \gamma_n^{(1)}$, we have

$$\Pr\left\{\exists f \in \mathcal{F}_{\succeq \gamma_n, \varepsilon} : \mathsf{P}_n f \leq \mathsf{P}_n f^* + \frac{\gamma_n}{2}\right\} \leq \frac{\delta}{2}.$$

Next, we control the behavior of the subclass $\mathcal{F}_{\succeq \gamma_n} \setminus \mathcal{F}_{\succeq \gamma_n, \varepsilon}$. From Theorem 6, if $\delta \leq \frac{1}{2}$

$$\Pr\left\{\exists f \in \mathcal{F}_{\succeq \gamma_n} : \mathsf{P}_n f < \mathsf{P}_n \pi(f) - \gamma_n^{(2)}\right\} \leq \frac{\delta}{2}.$$

for $\gamma_n^{(2)} = \frac{V}{n} \left(1080\mathcal{C} \log(2Kn) + 90\sqrt{\left(\log \frac{2}{\delta}\right) \mathcal{C} \log(2Kn)} + \log \frac{2e}{\delta}\right)$.

Now, combining the above two high probability guarantees, with probability at least $1 - \delta$ both statements below hold for all $f \in \mathcal{F}_{\succeq \gamma_n}$:

$$\mathsf{P}_n f \geq \mathsf{P}_n \pi(f) - \gamma_n^{(2)}$$
$$\mathsf{P}_n \pi(f) \geq \mathsf{P}_n f^* + \frac{\gamma_n}{2}.$$

Thus, with the same probability, for all $f \in \mathcal{F}_{\succeq \gamma_n}$:

$$\mathsf{P}_n f \geq \mathsf{P}_n f^* + \frac{\gamma_n}{2} - \gamma_n^{(2)}.$$

Setting $\gamma_n = (\gamma_n^{(1)} \vee 2\gamma_n^{(2)}) + \frac{1}{n}$, and recalling that ERM selects hypotheses purely based on their empirical risk, we see that with probability at least $1 - \delta$, ERM will not select any hypothesis whose excess risk is at least

$$(\gamma_n^{(1)} \vee (2\gamma_n^{(2)})) + \frac{1}{n}. \qquad \square$$

## Footnotes

[2] We scale by $\eta$ here because we are chasing a certain $\eta$-dependent rate.

[3]Boucheron et al. [5] stated this result in terms of packing numbers, but careful inspection of their proof reveals that the argument works for covering numbers as well. Moreover, other proofs generally use covering numbers.

[4] It may be possible to get a weaker dependence on $V$ with a more careful argument that depends on $V$ throughout; in particular, Talagrand's inequality can handle the parameter $V$.