[Reviews · NeurIPS 2014]

Submitted by Assigned_Reviewer_17

This paper shows how to obtain fast rates for ERM from a very general assumption known as "stochastic mixability".
The paper is clear, well-written, well-motivated, well-referenced, and presents a solid technical contribution.

Specific comments:
- in Sec. 2.1: the loss is assumed to be Lipschitz, so presumably this would not apply to classification problems with 0-1 loss.
It should be made more clear that only "regression"-type learning problems are being considered.
Also, F is stipulated as being compact, yet no metric/norm/topology is specified. An expert will infer that the norm in question
is L_1(P) --- which is vindicated by the covering numbers defined in Sec. 4. But the topology should be specified before compactness
is mentioned.

- in Sec. 2.4, there is some inconsistency of notation between E Z_f and P Z_f. Either is fine, but please be consistent (or point
out the difference if the two are distinct).

Finally, on the issue of bounded losses, which the authors bring up as an assumption to relax in the future, please take a look at

A. Kontorovich: Concentration in unbounded metric spaces and algorithmic stability, ICML 2014

and the numerous references therein.
Summary: An interesting, solid contribution, worthy of publication.

Submitted by Assigned_Reviewer_31

This paper shows that the statistical condition of stochastic mixability of a loss class is sufficient to proving fast rates of convergence of the ERM on this class to the oracle loss, for parametric and nonparametric classes. Further it is shown that lack of stochastic mixability implies non-unique minimizers, which in some cases is known to lead to slow rates.
The main contribution here is pointing out the usefulness and generality of stochastic mixability as a condition for fast rates. The bounds themselves are not optimal, as noted by the authors, but might be improved using localization techniques.

The observation this work provides is interesting, and might be useful in further characterizing loss classes that obtain fast rates versus those that do not. It would have been beneficial if a comparison between the new results and their previous counterparts, which rely on different conditions on the loss class. Especially in the parametric case, where localization might not be necessary, this could shed more light on the generality of the proposed approach.
Summary: An interesting observation on the usefulness of stochastic mixability as a condition for fast rates. A direct comparison to similar results that rely on other conditions would have been helpful.

Submitted by Assigned_Reviewer_40

This paper considers the convergence rate of excess risk for empirical risk minimization. In particular, the authors sharply characterize the slow and fast rate phenomenon using the stochastic mixability of the loss function. In detail, it is shown that stochastic mixability implies fast rates for VC-type classes.

I think this paper is very well-written and presented. The authors successfully delivered their insightful observation.
Summary: This paper is very well-written and presented. The authors successfully delivered their insightful observation. It is certainly among the top 50% of the accepted NIPS papers.
Author Feedback
Author rebuttal: Thanks to all the reviewers for your constructive comments on our submission.

Since submission we have discovered a flaw in our proof of fast rates for VC-type classes (Theorem 5). However, we can still prove a slightly weaker result for classes with polynomial uniform L1 covering numbers with bracketing. We now explain this problem and summarize how we can prove the slightly weaker result.

The argument in the paper begins by establishing that empirical risk minimization (ERM) will not pick a more than ~ 1/n bad hypothesis. It relies on this hypothesis being a fixed function not depending on the training sample. This part of the paper is still correct. The problem is in how we applied Theorem 1 during the proof of Theorem 5.

Theorem 5 involves taking a union bound over a certain set of bad functions (i.e. worse than ~ 1/n bad functions), and applying Theorem 1 to each of those bad functions, to show that with high probability ERM will not pick any of them. This would be fine if the set of functions were fixed, but these functions were chosen to be elements of an epsilon-net in the L1(P_z) norm (the empirical L1 norm); this can be seen from the first line of the proof of Theorem 5. Consequently, each function of this epsilon-net is not independent of the training sample, and so Theorem 1 cannot be applied in the way we did.

We have since been able to prove a slightly weaker version of Theorem 5, where rather than assuming polynomial uniform L1 covering numbers we instead use the slightly stronger assumption of polynomial uniform L1 covering numbers with bracketing. This enables application to Theorem 5 for finite classes (based on a deterministic epsilon-net in the L1(P)-norm) along with an argument that shows, for each element g of the epsilon-net, with high probability the empirical risks of those elements covered by g are much smaller than the empirical risk of g (where the gap is of order 1/n). Hence, the finite classes result which gives the fast rate for elements of the epsilon-net, together with this local result, provide the fast rate for the entire class. It appears that the key tool we use in this local analysis also works for VC-classes (classes with finite VC dimension), but we are not claiming a rigorous proof of this yet. It may also work with VC-type classes (classes with polynomial uniform L1 entropy), but this requires more exploration.

Even with the proposed weakening, we think there is value in the geometric nature of our approach to concentration via the general moment problem. Also, the geometric understanding we build between the uniqueness of the minimizer and stochastic mixability seems interesting in itself.

All the results prior to Theorem 5 are still correct and would stay in the paper. Section 5, "Characterizing convexity from the perspective of risk minimization", is also still correct and would stay in the paper. However, we would plan to cut Section 6, "Weak stochastic mixability and nonparametric classes,” as this result builds most sensibly off of the original version of Theorem 5.

While the majority of the paper is still correct as submitted, and while it does provide a new approach to a core component of analyzing learning problems for fast rates (the reduction to the general moment problem and the connection with the geometry of multiple minimizers for general loss functions), we understand and accept that the identified bug and our proposed weakening of Theorem 5 may not be acceptable and we understand reviewers will want to reassess their positive view of the paper.